

# Resumption of donor-origin spermatogenesis in senescent goldfish *Carassius auratus* (Linnaeus, 1758) following spermatogonial cell therapy

Sullip K. Majhi, Mog Chowdhury, Santosh Kumar, Rajeev K. Singh, Vindhya Mohindra and Kuldeep K. Lal

National Bureau of Fish Genetic Resources, Lucknow, India

## ABSTRACT

Stem cell research has come into prominence because of its applications in assisted reproductive technology and the treatment of deadly diseases. In teleost fishes, spermatogonial stem cells have been effectively used to produce surrogate gametes and progeny through germ cell transplantation technique. The present study is the first report of an innovative application of stem cell therapy in fish species for revitalising the reproductive competence of senescent individuals. Senescent male goldfish, *Carassius auratus*, approximately 10 years of age were procured from a fish-breeding farm and were reared locally in the lab for an additional two years. The senescence of the individuals was then evaluated and confirmed using histological analysis, gonadal index assessment, and germ-cell specific *vasa* gene expression. Analyses revealed absence of spermatogonial cells and other germ cells in the testes of the senescent fish ($n = 5$). Spermatogonial cells from sexually immature *C. auratus* male donor were isolated using discontinuous percoll gradients, labelled with the fluorescent dye PKH-26, and transplanted into the gonads of senescent *C. auratus* males through urogenital papilla. Six months after the transplant, spermatozoa were collected through applying gentle manual pressure on the abdomen and were observed under a microscope. All *C. auratus* males with the transplant had produced spermatozoa from the transplanted cells. This was confirmed by the retention of PKH-26 in the spermatozoa and diagnostic SSR locus. Gravid *C. auratus* females were artificially inseminated with the spermatozoa of those senescent males and natural spawning was allowed. As a result viable progeny were produced. These observation suggests that the reproductive competence of senescent male fishes can be revitalised through spermatogonial stem cell therapy to produce functional gametes.

# INTRODUCTION

Stem cells are described as undifferentiated cells that have the potential to renew themselves and differentiate into a single cell type or multiple specialised cell types as they can undergo numerous cycles of cell division while maintaining their undifferentiated state (*Takeuchi, Yoshizaki & Takeuchi, 2004*). Currently, stem cell research is one of the upcoming areas of

Corresponding author
Sullip K. Majhi, skmajhi@nbfgr.res.in

research in science. Since their discovery and the subsequent establishment of protocols for their successful isolation and culture, researchers have used them for various purposes (*Zhu et al., 2010*), including treatment of chronic diseases and reproductive ailments (*Nazari et al., 2015*). Stem cell therapy has also shown the potential to revitalise or repair malfunctioning organs by using unscathed donor cells, which can take repair the damaged organ through their potential of differentiation to make the organ function normally and provide animals with an extended and improved life (*Min-Wen, Jun-Hao & Shyh-Chang, 2016*). In humans, this has opened a new avenue for regenerative medical approaches and has become a viable treatment option in healing chronically damaged organs (*Zhao et al., 2014*). However, in teleost fish, the use of stem cells for therapeutic purposes has not been explored so far; nevertheless, this therapy has the potential to revitalise malfunctioning reproductive organs of both sexually incompetent or senescent fishes that have a high commercial value or have become critically endangered. Currently, in fish, stem cells are widely used for surrogate brood stock development through germ-cell transplantation (*Majhi et al., 2009*; *Lacerda et al., 2015*; *Majhi et al., 2014*). In this case, the donor cells obtained from young hatchlings or adults are transplanted into the recipient fish at various developmental stages, and on maturation, the recipient fish produce donor-derived gametes (*Majhi et al., 2014*; *Takeuchi, Yoshizaki & Takeuchi, 2003*). In this study, we demonstrate successful restoration of spermatogenesis in senescent *C. auratus* through intragonadal stem cell therapy for the first time. The simple approach described in this study ultimately leads to generation of viable functional spermatozoa in senescent *C. auratus* males, which can fertilise eggs derived from young *C. auratus* females and produce healthy progeny. Thus, this technique extends the reproductive lifespan of the fish beyond the pubertal phase and effectively generates viable progeny.

## MATERIAL AND METHODS

### Ethics statement

This study was approved by the Animal Ethics Committee of National Bureau of Fish Genetic Resources (#G/CPCSEA/IAEC/2015/2).

### Animals, age, and rearing protocols

Male goldfish (*C. auratus*; $n = 60$; mean body weight $\pm$ standard deviation [SD] of $230 \pm 20.5$ g) were procured from Maharashtra, India, which were reported to be approximately 10 years old by the providers. The age of the fish was evaluated and confirmed by counting the growth rings located at scale (the growth rings from the fish raised in the farm with known age was used as reference; Fig. S1). The fish were stocked in 500 L tanks after procuring them from the farm at the density of 5.0 kg of fish per m³ and reared in flow-through fresh water system (temperature: 25 °C $\pm$ 2 °C; dissolved oxygen: 5.3–6.1 ppm; pH: 7.5–8.0; hardness: 40–45 ppm) under a constant light cycle (12 h light and 12 h dark). The fish were reared for an additional two years before confirmation of their senility through histological analysis and germ-cell-specific *vasa* gene expression studies. The prepubertal donor *C. auratus* males (three months old) were produced at the

rearing facilities of the National Bureau of Fish Genetic Resources, Lucknow. Both groups of animals were fed a commercially available pelleted diet twice a day to satiation.

## Gonad histology

After two years of rearing, five fish were randomly sampled and sacrificed using an overdose of anaesthesia (2-phenoxyethanol; HiMedia, India), and their body weights were recorded. The gonads of the fish were excised and weighed, macroscopically examined, and photographed using a digital camera. The middle portion of the right and left gonads from each fish were then immersed in Bouin's fixative for 24 h and preserved in 70% ethanol. The gonads were processed for microscopic examination following routine histological procedures up to the stage of preparing 5-$\mu$m thick cross-sections and staining them with hematoxylin–eosin. Histological sections from each fish were examined under a microscope at magnifications between 10$\times$ and 60$\times$.

## Gene expression analysis

Samples for real-time reverse transcription polymerase chain reaction (RT-PCR) of *vasa* gene expression were obtained from the anterior region of the testes after the 2-year rearing period ($n = 3$) and at six months after spermatogonial cell therapy ($n = 3$). The samples were stored in RNA*later* (Sigma–Aldrich, St.Louis, MO, USA) at $-80\ °C$ until further processing. RNA was extracted using TRIzol (Invitrogen Life Technology, Carlsbad, CA, USA) according to the manufacturer's protocol. cDNA was synthesised using a first-strand cDNA synthesis kit (Thermoscientific, USA). The primers for real-time RT-PCR were (5′-AACCCTCATGTTCAGCGCCAC-3′ and 5′-TGGTTTCAACAAAGACCATCGTGC-3′). The real-time PCRs were run in an ABI PRISM 7300 (Applied Biosystems, USA) system using *Power* SYBR® Green PCR Master Mix in a total volume of 15 $\mu$L, which included 7.5 $\mu$L of 2 $\times$ Maxima$^{TM}$ SYBR Green qPCR Master Mix (Thermoscientific, USA), 20 ng of first-strand cDNA and 5 pmol L$^{-1}$ of each primer. $\beta$-actin (5′-GAC TTC GAG CAG GAG ATG G-3′ and 5′-CAA GAA GGA TGG CTG GAA CA-3′) was used as an endogenous control. The comparative Ct method was used for *vasa* mRNA quantification. The fold change in the expression level was calculated using the $2 - \Delta\Delta$Ct method (*Whelan, Russell & Whelan, 2003*).

## Isolation and labelling of donor cells

The donor *C. auratus* males ($n = 3$) were sacrificed using an anaesthetic overdose, and their testes were excised and rinsed in phosphate buffered saline (PBS; pH = 8.2). The testicular tissue was finely minced and incubated in a dissociating solution containing 0.5% trypsin (pH 8.2; Worthington Biochemical Corp., Lakewood, NJ), 5% foetal bovine serum (JRH Biosciences, Lenexa, KS), and 1mmol L$^{-1}$ Ca$^{2+}$ in PBS (pH 8.2) for 2 h at 22 °C. The dispersed testicular cells were sieved through a nylon screen (mesh size 50 $\mu$m) to eliminate the non-dissociated cell clumps, suspended in discontinuous percoll (Sigma–Aldrich, St. Louis, MO, USA) gradients of 50%, 25%, and 12% and centrifuged at 200 $\times$ $g$ for 20 min at 20 °C (*Majhi et al., 2014*). The bottom phase containing predominantly spermatogonial cells (determined during preliminary trials which involved cell size measurement) was harvested, and the cells were subjected to rinsing as well as a cell viability test through

trypan blue (0.4% w/v) exclusion assay. The cells were then exposed to the PKH-26 Cell Linker (Sigma–Aldrich, St.Louis, MO, USA) at the concentration of 8 µmol/mL (room temperature, 10 min) to label the cells for tracking their behaviour inside the senescent (recipient) *C. auratus* gonads. The staining procedure was stopped by the addition of an equal volume of heat-inactivated foetal bovine serum. The labelled cells were rinsed three times to remove the unincorporated dye, suspended in Dulbecco Modified Eagle Medium (Life Technologies, Rockville, MD) with 10% foetal bovine serum, and placed on ice until transplantation.

## Cell therapy procedure

Forty fishes were first anaesthetised using 200 ppm phenoxyethanol and placed on a cell transplantation platform, where they received a constant flow of oxygenated water containing 100 ppm of the anaesthetic drug. To prevent desiccation, the surface of the fish was kept moisturised during the entire procedure of cell transplantation, which lasted for approximately 5–7 min per fish on average. A micro syringe was used to inject the cell suspension into the testicular lobe through genital papilla (Fig. S2). Each fish was injected with 100 µL of a cell suspension containing approximately $5 \times 10^4$ cells, at the flow rate of approximately 20 µL/min. Trypan blue was added to the injection medium prior to transplantation to facilitate visualisation of the cell suspension inside the needle and inside the gonads after injection. The genital papilla region was topically treated with 10% isodine solution and the fish were returned to clean water.

## Analysis of donor cells post therapy

The fate of the donor cells after transplantation was analysed through fluorescent microscopy at six and 12 weeks after injection. Then, the testes from the five animals chosen randomly during each sampling were removed, washed in PBS (pH 8.2), macroscopically examined for the degree of dispersion of the cell suspension (Fig. 1), and immediately frozen in liquid nitrogen. Cryostat (Leica CM 1500, Germany) section 8 µm thick were cut from the representative portions of these testes and observed under a fluorescent microscope (Nikon Eclipse E600, Tokyo, Japan) for detecting the presence of PKH-26-labelled donor cells.

The fate of the donor cells was then examined by measuring the gonado-somatic index (GSI; $n = 5$) and sperm density at six months after therapy. On each occasion, 20–30 µL of sperm was collected from each of the 25 therapy-treated males and five control males. Sperm was manually extracted by applying gentle abdominal pressure after careful removal of urine and wiping the genital papilla. The sperm samples (10 µL) were then diluted 1,000 times with PBS and the density of spermatozoa was counted using a haemocytometer (Marienfeld, Germany) under a microscope. Some of the spreads of sperm used for counting were also observed under the fluorescent microscope for the detection of PKH-26-labelled cells.

Sperm and blood samples were collected at 18 months from the spermatogonia cell transplanted senescent male ($n = 25$). Total DNA was extracted using PureLink Genomic DNA kit (Invitrogen Life Technology, Carlsbad, CA, USA) according

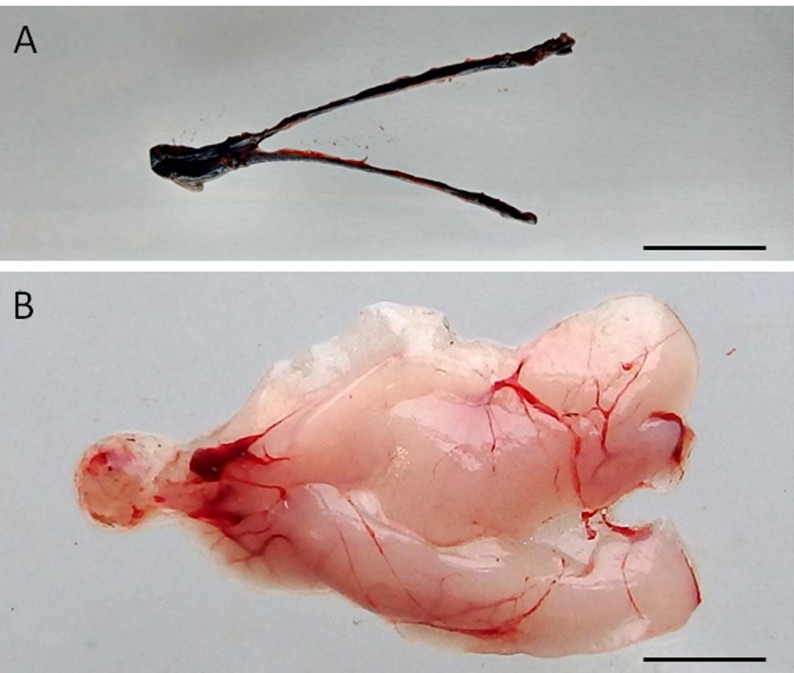

**Figure 1** **Visualization of the dispersal of the cell suspension through the gonad of senescence *C. auratus* after transplantation.** (A) Macroscopic appearance of the testis 48 hours after spermatogonia cell transplantation (note the diffusion of the marker Trypan blue throughout the testis). (B) Appearance of the testis 6 months after cell transplantation (note the transplanted cells in senescence testis have undergone proliferation and differentiation to match the testis of a sexually mature individual). Scale bars indicate 2 cm.

to the manufacturer's protocol and DNA typing was done using *C. auratus* SSR locus *J60* * (forward 5′-CTGGCTGTCTGATCCTGCTGAT-3′ and reverse 5′-TGGCCAGAGTTTAAAAACCAGTCC-3′) (*Yue & Orban, 2002*). PCR reaction was performed in 25 µL consisted of 1x Taq buffer, 1.5 mM $MgCl_2$, 0.5 µm each primer, 200 µM of dNTP, 0.75 U Taq DNA polymerase and 80 ng of the DNA template. The amplification was done in a thermal Cycler (ThermoFisher Scientific, USA) and consisted of an initial denaturation at 94 °C for 5 min, 25 cycles of 94 °C for 30 sec, 62 °C for 30 sec and 72 °C for 30 sec, followed by elongation at 72 °C for 10 min. Amplified products was visualized by 10% polyacrylamide gel electrophoresis (PAGE) and silver staining. Amplicon sizing was done on gel imaging and analysis system (UVP) using Msp I digested pBR322 as ladder.

## Artificial fertilization and induced spawning

After 6 months, the senescent therapy-treated *C. auratus* males ($n = 25$) resumed production of spermatozoa. Artificial insemination and natural spawning were performed using eggs from wild *C. auratus* females. Approximately 20 µL of milt from each of the ten therapy-treated males was used to fertilise a batch of *C. auratus* eggs. The batches of eggs were then incubated under flowing water at 25 °C until hatching. In the natural spawning trials, ten senescent therapy-treated males and wild females were paired in a

100 L glass aquarium and reared in fresh water (Temperature: 25 °C ± 2 °C; dissolved oxygen: 5.3–6.1 ppm; pH: 7.5–8.0; hardness: 40–45 ppm) under a constant light cycle (12 h light and 12 h darkness). Every, morning between 10:00 and 11:00 h, the tanks were examined for spawning and embryos were collected. The fertilised eggs obtained from each cross were incubated at 25 °C and observed under a light microscope for the assessment of fertilisation, embryonic development, and hatching.

## Statistical analysis

The statistical significance of the differences in *vasa* gene expression levels, GSI values and sperm densities among the groups is evaluated using one-way analysis of variance and Tukey's multiple comparison test. Graphpad Prism ver. 4.00 (Graphpad Software, San Diego, Carlifornia, USA) is used for statistical analyses. Data are presented as mean ± SD and the differences among the groups are considered as statistically significant at $p < 0.05$.

# RESULTS

## Histological observation of germ cells

Microscopic examination of the testes revealed that all the five senescent *C. auratus* males aged 12 years exhibited shrunken gonads (Fig. 1A) and complete disappearance of all the stages of germ cells (Figs. 2A–2B) in all sections examined before stem cell transplantation. By contrast, the control males (age >1 year) exhibited active spermatogenesis with a large cyst of spermatogonial cells, all stages of germ cells (Figs. 2C–2D) and their efferent duct contained spermatozoa. These histological observations were also corroborated by the results of GSI evaluation and real-time RT-PCR analysis, which showed that both GSI (Fig. 3) values and *vasa* gene transcript levels (Fig. 4) were significantly lower in the senescent males compared to the controls ($p < 0.05$).

## Fate of transplanted cells in senescent testes

The transplanted donor spermatogonial cells were found randomly distributed throughout the spermatogenic lobules in all the five senescent males examined 6 weeks after the therapy. At 12 weeks after the therapy, the donor spermatogonial stem cells had reached the blind end of the lobules (cortical region of the testis; Figs. 5A–5D) and had undergone proliferation to form a network along the testicular lobule; this stage was observed in all the sampled fish ($n = 5$; Fig. 5E). At six months after the therapy, the sperm could be collected by applying gentle pressure on the abdomen of all 25 senescent *C. auratus* males (Fig. 5F). The sperm density, which was not detectable before the therapy, had significantly increased after the therapy and was comparable with that of the reproductive control males (Fig. 6).

## Production of viable gametes and progeny from senescent males

Six months after cell therapy, the senescent *C. auratus* males resumed spermatogenesis and produced spermatozoa from the transplanted cells; the origin of the spermatozoa (from the transplanted cells) was confirmed by the existence of two different genotypes and presence of the red fluorescent labels in the cells (Fig. 5F). Locus *J60\** differentiated amplicons from blood (recipient) and sperm (donor cell) DNA (Figs. 7 and 8). These

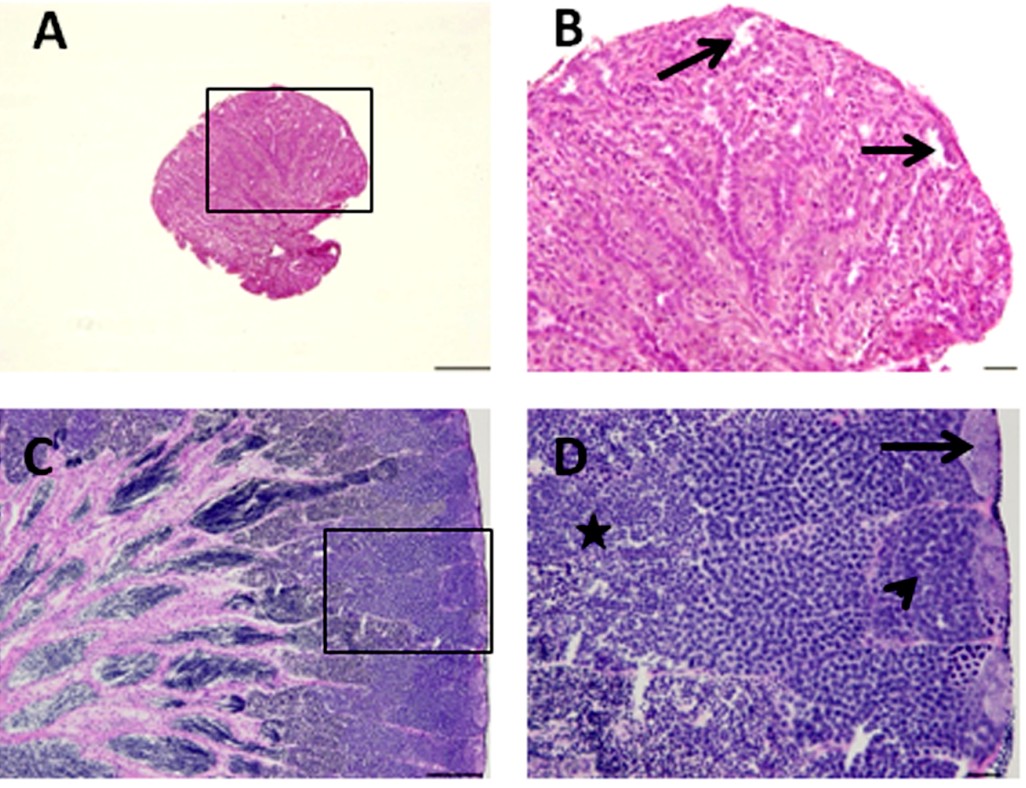

**Figure 2 Histological appearance of senescence and control *C. auratus* gonads.** (B) and (D) are high magnifications of insets of (A) and (B). (A, B) Twelve year old senescence testis showing empty niche (arrows) with complete absence of spermatogonia and other germ cells. (C, D) A sexually mature control testis indicating large cysts of spermatogonia in the blind end of the spermatogenic lobules (arrows), spermatocytes (arrowhead) and spermatids (star) depicting an active spermatogenesis within the lobules. Scale bars indicate 100 μm (A and C) and 20 μm (B and D).

therapy-treated males were then used for artificial fertilization and for trials of induced spawning of eggs obtained from young *C. auratus* females (Tables 1–2). These crosses resulted in normal embryonic development and hatching, which were similar to those in the control fish. Breeding between the spermatozoa from therapy-treated males and eggs from wild *C. auratus* females resulted in 88.6%–97.5% hatching whereas 93.8%-97.7% result was obtained in the control (Table 1). On the other hand, when the senescent males were coupled with wild *C. auratus* females for induced spawning, the breeding resulted in 95.5%–99.5% hatching with normal embryonic development and viable progeny; oppose to 97.7%–98.4% in control (Table 2). These observations suggest the viability of the proposed approach in revitalising the reproductive competence of the males from commercially valuable fish species that have aged and can no more impregnate the females.

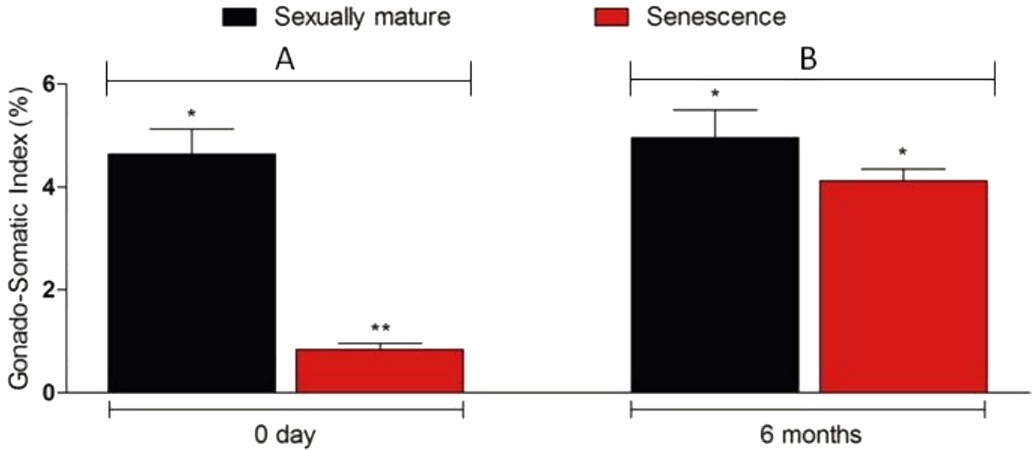

**Figure 3 Changes in the gonado-somatic index of senescence and control gonads between 0 day (A) and six months (B).** Six months after therapy the gonado-somatic index value of senescence *C. auratus* males have significantly increased to match the sexually mature control. Data are presented as mean ± SD. Columns with double asterisk vary significantly (ANOVA—Tukey test, $p < 0.05$).

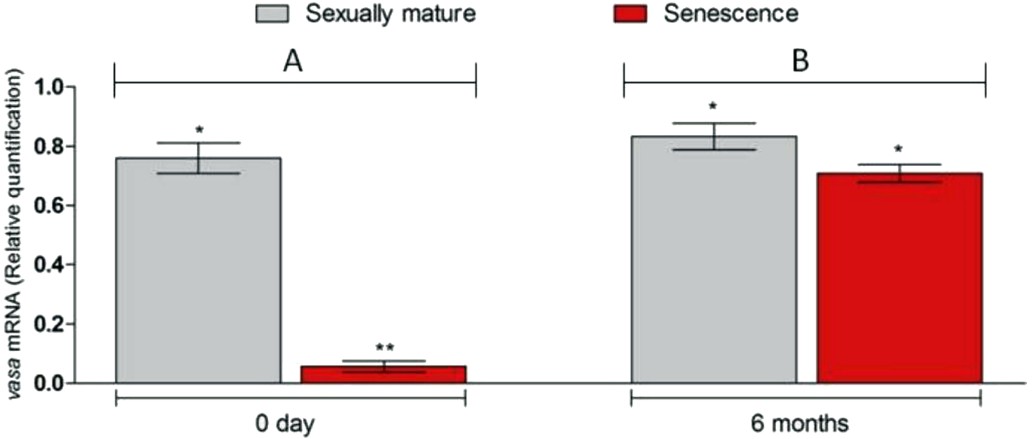

**Figure 4 Changes in *vasa*. gene transcript levels in senescence and control *C. auratus* gonads between 0 day (A) and six months (B).** Six months after therapy the *vasa* gene transcript levels in senescence *C. auratus* males have significantly increased to match the reproductive control. Data are presented as mean ± SE. Columns with double asterisk vary significantly (ANOVA—Tukey test, $p < 0.05$).

## DISCUSSION

In the present study, spermatogonial cells harvested from prepubertal *C. auratus* testes were successfully transplanted for restoring the reproductive competence of senescent *C. auratus* males by using cell-therapy. As observed in mammals (*Brinster & Zimmermann, 1994*; *Ogawa, Dobrinski & Brinster, 1999*), the therapy resulted in recolonization of the seminiferous epithelium, resumption of spermatogenesis, and production of functional spermatozoa. The simple but effective procedure demonstrated in this study can be immediately applied in hatcheries and other seed production facilities. This procedure has

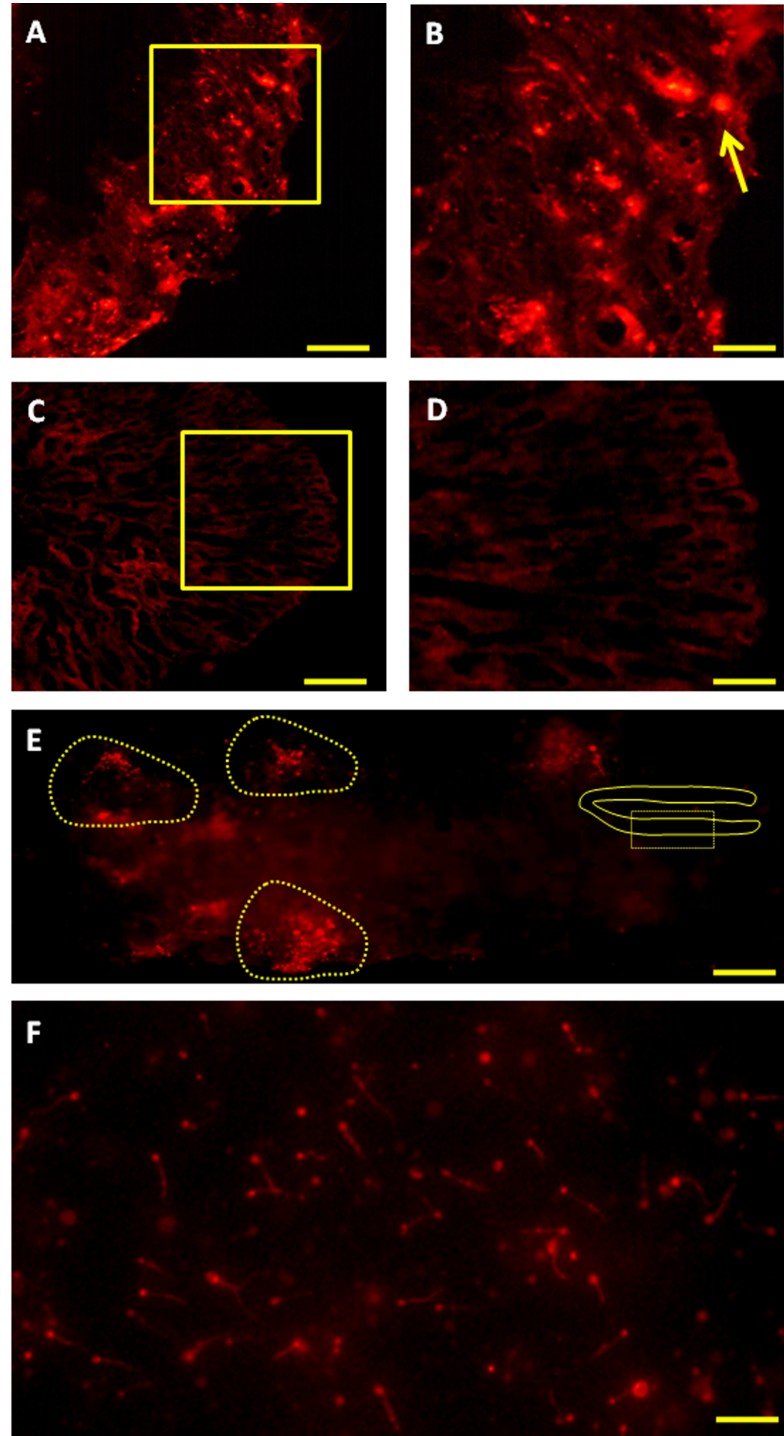

**Figure 5** **Fate of PKH-26-labeled donor spermatogonia cells in senescent *C. auratus* males examined between four weeks and six months after transplantation.** (A, B) Cryostat section of a transplanted testis at four weeks showing the presence of transplanted spermatogonia cell at the blind end of the spermatogenic lobules (arrow; B is a high magnification of the box in A). (C, D) Cryostat section of a non-transplanted, control testis at four weeks showing the approximate location of the blind end of the spermatogenic lobules (D is a high magnification of the box in C). 

**Figure 5 (...continued)**
(E) Whole-mount preparation of a transplanted testis at twelve weeks showing the proliferation of donor-derived cells (highlighted) along the length of the gonad. (F) Whole-mount preparation of spermatozoa cell derived from the senescent male 6 months after the procedure (note that the spermatozoa produced were of donor-origin that was characterized by retention of PKH-26 dye). Scale bars indicate 100 µm (A and C), 20 µm (B and D), 500 µm (E) and 50 µm (F).

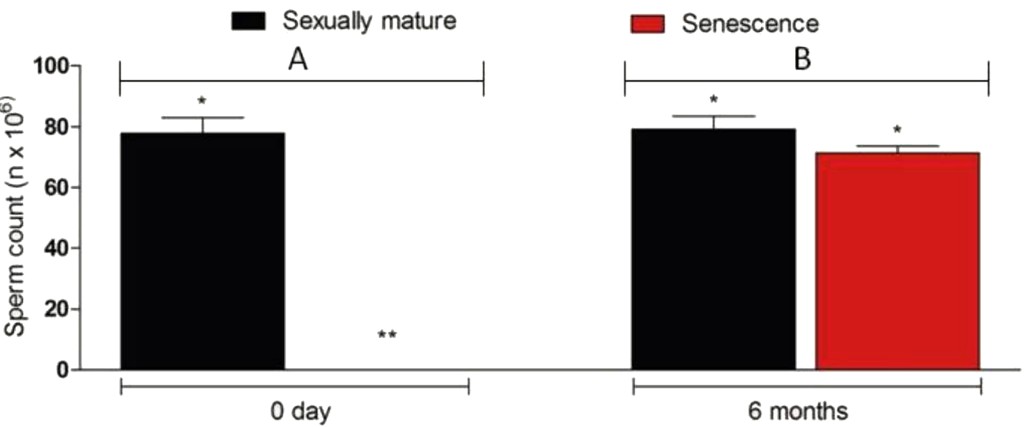

**Figure 6** **Sperm density in senescent recipients and non-transplanted (negative control) *C. auratus* between 0 day (A) and 6 months (B) after spermatogonial cell therapy.** The sperm count in the senescent males had significantly increased after spermatogonial cell therapy. Data are presented as mean ± SD. Columns with double asterisk vary significantly (Tukey's multiple comparison test, $p < 0.05$).

a considerable potential for extending the reproductive phase of commercially valuable species.

The testes recovered from the senescent *C. auratus* males exhibited shrinkage of testicular lobes, absence of germ cells, and deposition of adipose and connective tissue. The germinal epithelium was continuous, and the testicular cross-section was noticeably smaller than that observed in reproductive males. Notably, this observation is contrary to that in senescent guppy (*Lepistes reticulatus*) in which histological changes in the testes revealed a progressive increase in the percentage of lobules containing spermatids and spermatozoa and an increase in the deposition of melanin as well as adipose and connective tissue (*Woodhead & Elett, 1969*). However, in senescent mosquito fish (*Gambusia affinis*) and European bitterling (*Rhodeus amarus*), partial testicular degeneration was observed to occur with a considerable reduction in the number of germ cells (*Woodhead, 1979*; *Rasquin & Hafter, 1951*; *Haranghy et al., 1977*). According to *Finch (1990)* such variations in the gonadal germ cell profiles and testicular morphologies of senescent fishes are due to their growth patterns and origins of habitation. Thus, the characteristics of senescence in fishes cannot be generalised and vary widely depending on their habitation history and other physico-chemical parameters of their ecosystem (*Liu & Walford, 1966*). For instance, tropical fish species grow faster and attain senescence comparatively earlier than do temperate fish species (*Patnaik, Mahapatro & Jena, 1994*). The experimental fish used in this study were reared in a tropical environment (temperature range: 25 °C–32 °C) for

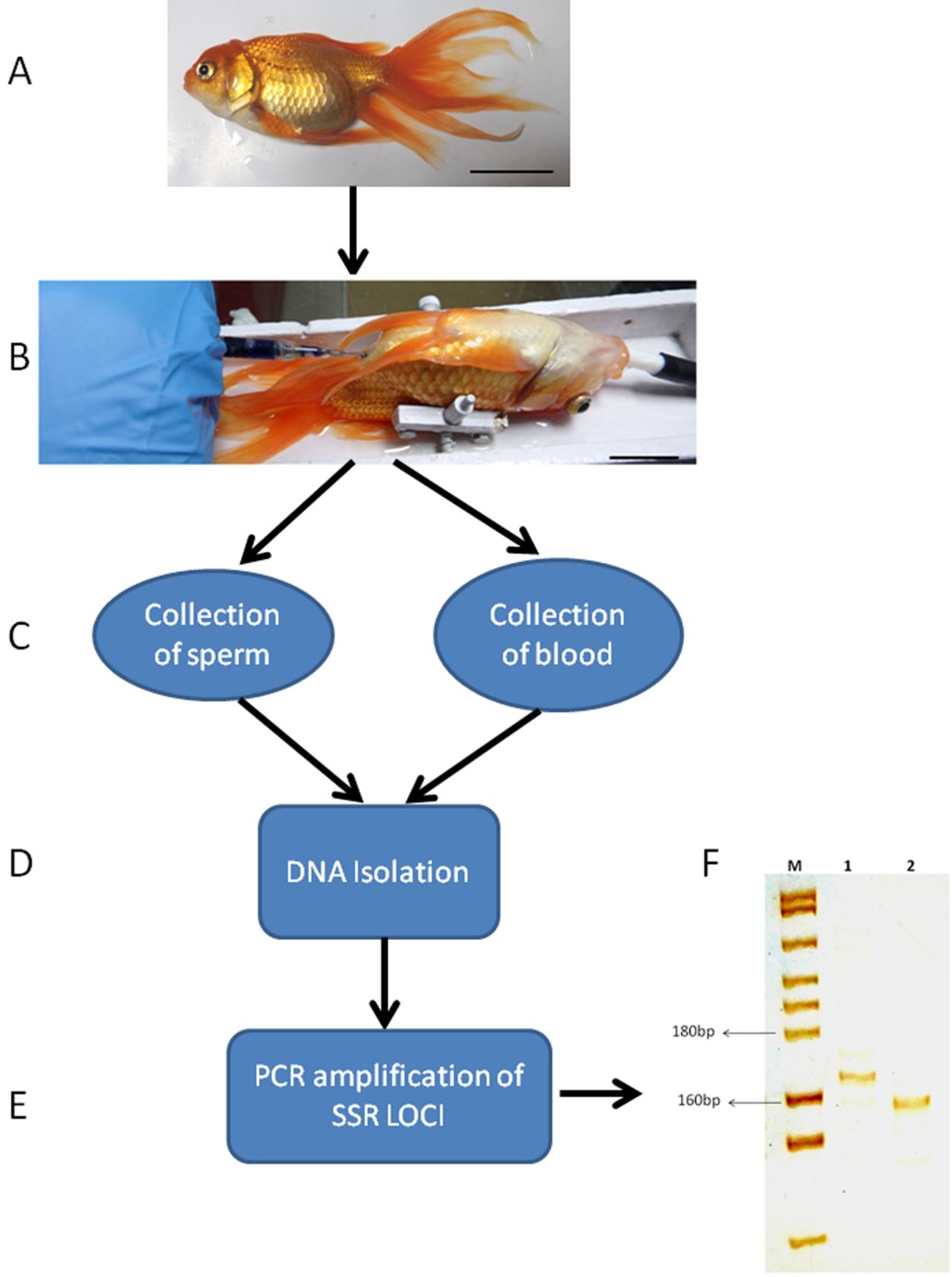

**Figure 7** **Genotyping of sperm and blood collected from a senescent *C. auratus* males 18 months after the cell transplantation.** (A) The senescent *C. auratus* male used in the experiment. (B) The donor spermatogonial cell transplantation in senescent *C. auratus* male. (C) Collection of sperm and blood samples from senescent *C. auratus* males 18 months after the cell transplantation. (D) DNA isolation from the samples using PureLink Genomic DNA kit according to the manufacturer's protocol. (E) The microsatellite primers were used for amplification of DNA for genotyping. (F) Lanes include molecular marker (M), blood of senescent male (1) and sperm (2). Note that the donor-derived spermatozoa were detected in the sperm of senescent recipient shown in lane 2.

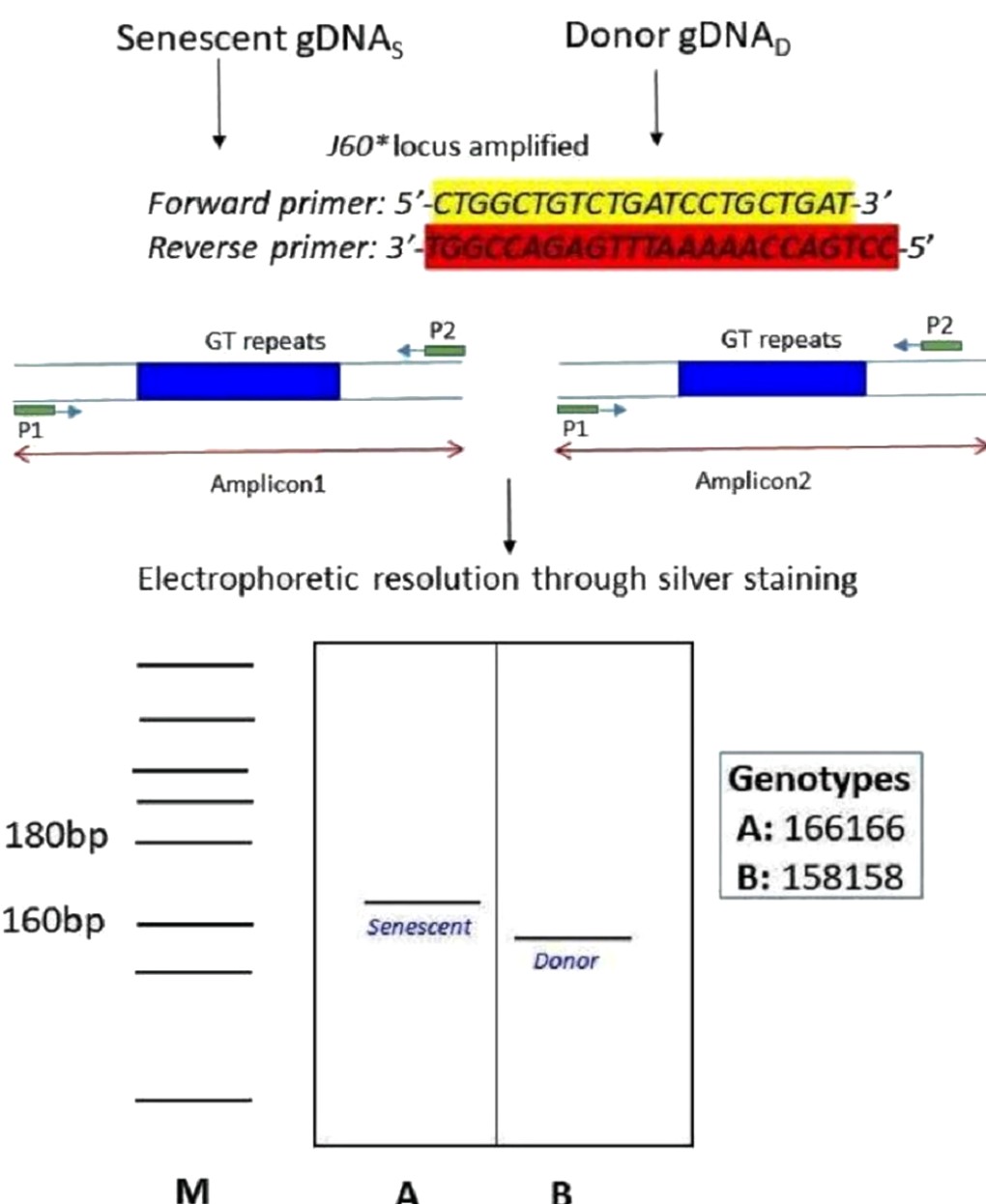

**Figure 8** Schematic illustration of species-specific microsatellite loci J60* amplifying both senescent (recipient) and donor genomic DNA. The species-specific microsatellite loci J60* (*Yue & Orban, 2002*) amplifying both senescent (recipient) and donor genomic DNA and visualization through silver staining. The two genotypes (A) senescent 166/166 and (B) donor (158/158) were electrophoretically resolved on polyacrylamide gel using Msp I digested pBR322 as ladder (M).

10 years; probably this may explain why germ cell degeneration in the senescent individuals was considerably more severe than that previously reported in other fish species.

Regardless of the differences between sterile (natural or experimentally induced) and senescent testes, the process of recolonization by the transplanted spermatogonial cells was similar to the previously reported process (*Majhi et al., 2009*; *Lacerda et al., 2015*; *Majhi*

**Table 1 Results of artificial fertilization.** The eggs derived from wild gravid *C. auratus* females with sperm from ten senescence *C. auratus* males ( $n = 10$; transplanted with spermatogonia derived from pre-pubertal *C. auratus* donors) and control (crosses between sexually mature *C. auratus* males and females). Data presented are mean $\pm$ SD.

| Fertilization trail | Eggs (*n*) from wild gravid *C. auratus* females | Fertilization rate (*n*; %) | Hatching rate (*n*; %) |
|---|---|---|---|
| senescence *C. auratus* males ($n = 10$) | $476 \pm 107$ | $453 \pm 112$ | $432 \pm 109$ |
| Control ($n = 3$) | $503 \pm 99$ | $476 \pm 91$ | $460 \pm 95$ |

**Table 2 Results of induced spawning.** The cross between senescence *C. auratus* males ($n = 10$; transplanted with spermatogonia derived from pre-pubertal *C. auratus* donors) with wild gravid *C. auratus* females and a control (paired between sexually mature *C. auratus* males and females). Data presented are mean $\pm$ SD.

| Fertilization trail | Total number of embryo collected (*n*) | Fertilization rate $\pm$ (*n*; %) | Hatching rate $\pm$ (*n*; %) |
|---|---|---|---|
| senescence *C. auratus* males paired with gravid *C. auratus* females (n=10) | $1068 \pm 271$ | $1046 \pm 273$ | $1026 \pm 277$ |
| Control (n=3) | $1091 \pm 140$ | $1068 \pm 128$ | $1047 \pm 127$ |

*et al., 2014*; *Ogawa et al., 2000*). For instance, after cell therapy, many PKH-26-labelled spermatogonial cells eventually settled along the blind end of the seminiferous lobules and began proliferation within weeks of transplantation. Considering that only spermatogonial stem cells can migrate and settle at the basement membrane and resume the process of spermatogenesis (*Ogawa, 2001*), we can surmise that the cells used for therapy were spermatogonial stem cells. This conclusion is also proved by the fact that when the senescent *C. auratus* males (26 months after cell therapy) were repeatedly paired with wild *C. auratus* females for natural spawning, they continued to produce viable progeny to date. Transmembrane protein molecules present at the junctional complex located in the seminiferous tubules have been reported to transduce signals, maintain cell polarity, and mediate germ cell migration (*Wang & Cheng, 2007*). Although we did not examine the endocrine regulation involved in the migration, proliferation and differentiation of the donor spermatogonial cells inside the gonads of senescent *C. auratus* males, the results obtained in the present study (time-course observations of donor spermatogonial cells after therapy) indicated that the donor spermatogonial cells might have sensed and responded to the molecules released from the blind end of the seminiferous lobules in therapy-treated senescent *C. auratus* males. Consequently, these spermatogonial cells migrated and settled down near the basement membrane and resumed spermatogenesis. However, the duration for which the *C. auratus* males treated with spermatogonial cell therapy retain their reproductive competence and produce functional gametes remains to be determined.

The present study demonstrated that spermatozoa derived from therapy-treated senescent *C. auratus* males exhibited functional properties similar to those of the control spermatozoa in terms of fertilisation, embryonic development and hatching,

and no evidence of defective spermatozoa was observed. Defective spermatogeneis has been reported to occur most prominently when the donor and recipient animals are phylogenetically distant (*Nantel et al., 1996*). In this study, the spermatogonial cells derived from pre-pubertal *C. auratus* males were transplanted into senescent *C. auratus* males; thus, the transplanted cells were probably immunologically compatible with the gonadal environment of the senescent *C. auratus* males. Moreover, the progeny produced using the spermatozoa derived from therapy-treated *C. auratus* males exhibited a similar growth pattern to that of the progeny produced from the control males. These observations suggest that the reproductive competence of senescent *C. auratus* males could be successfully revitalised using spermatogonial cell therapy. Currently, we are examining the feasibility of the cell-therapy approach in revitalising the reproductive competence of female *C. auratus*.

In conclusion, the most remarkable achievement of this study is the production of functional spermatozoa from senescent *C. auratus* males after spermatogonial cells derived from the pre-pubertal *C. auratus* donor were transplanted into the testes of the senescent males through the genital papilla. This approach, which to the best of our knowledge has been validated for the first time in fish in this study, is a method to generate functional gametes for additional years after they have aged beyond reproductive capability. This is a crucial development in the breeding of rare and/or commercially valuable fish species that are developed as brooders and require sizeable investments in terms of feed and health care. Such fish species are used for only a few years by the hatcheries for seed production and are invariably discarded when they lose their reproductive competence because of age.

## ACKNOWLEDGEMENTS

The authors express sincere thanks to the Director, NBFGR, Lucknow for providing all necessary help to perform this work.

### Funding

This work was supported by a Grant-in-Aid from the National Bureau of Fish Genetic Resources (Institutional research grant # IXX10896). The funders had no role in study design, data collection and analysis, decision to publish, or preparation of the manuscript.

### Grant Disclosures

The following grant information was disclosed by the authors:
National Bureau of Fish Genetic Resources: IXX10896.

### Competing Interests

The authors declare there are no competing interests.

### Author Contributions

- Sullip K Majhi conceived and designed the experiments, performed the experiments, analyzed the data, prepared figures and/or tables, authored or reviewed drafts of the paper, procurement of experimental fish, and approved the final draft.

- Mog Chowdhury and Santosh Kumar performed the experiments, prepared figures and/or tables, and approved the final draft.
- Rajeev K. Singh and Vindhya Mohindra performed the experiments, analyzed the data, prepared figures and/or tables, and approved the final draft.
- Kuldeep K. Lal analyzed the data, authored or reviewed drafts of the paper, and approved the final draft.

## Animal Ethics

The following information was supplied relating to ethical approvals (i.e., approving body and any reference numbers):

This study was approved by the Animal Ethics Committee of National Bureau of Fish Genetic Resources (#G/CPCSEA/IAEC/2015/2).

## Data Availability

The raw data are available in the Supplemental Files.

## Supplemental Information

Supplemental information for this article can be found online at http://dx.doi.org/10.7717/peerj.9116#supplemental-information.

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
