# Peer review of "Resumption of donor-origin spermatogenesis in senescent goldfish *Carassius auratus* (Linnaeus, 1758) following spermatogonial cell therapy"

_PeerJ, doi:10.7717/peerj.9116_

## Round 0.1 · original submission · Major Revisions

Please optimize the written English in the help of native speaker. Furthermore, figures and tables should be changed and improved. Some figures could be merged and some more suggestions were given by the reviewers.

Reviewer 1 ·

Basic reporting

The English language should be improved to ensure that an international audience can clearly understand your text. Meanwhile, the format of the manuscript should be optimized.
The flowing is the minor issues in the manuscript.
Line 49: the year should be 2004.
Line 49-51: Please revise this sentence, because the meaning is similar to the sentence from Line 47-48.
Line 80: “500L” should be “500 L”
Line 123: “produce spermatozoa from the transplanted cells;” should be “produced spermatozoa from the transplanted cells;”
Line 134: “Forty fish” should be ‘Forty fishes”
Line 149: “8-µm” should be “8 µm”

Experimental design

No conmment

Validity of the findings

No comment

Additional comments

This manuscript reported that the spermatogonia stem cell therapy could extend reproductive lifespan of fish. These findings in fish are useful and meaningful. In the following, there are some major issues. The author should complete all these issues before acceptance.
Major issue:
1. The author should use “*” to indicate the significance.
2. Please optimize the written English in the help of native speaker.
3. The author should make a table to present all the tests of 12 years old fish and the fish after transplantation in figure 2 if they have the images.
4. The image A looks different from image C at the size in figure 3, whether they are at the same magnification.
5. Why the resolution of figure 4 was lower compared with others?
6. The author should draw a schematic diagram for the genotyping in figure 8.

·

Basic reporting

no comment

Experimental design

no comment

Validity of the findings

no comment

Additional comments

The study by Majhi SK et al reported the spermatogenesis in senescent goldfish by spermatogonia cell therapy. Overall, it is interesting and adds knowledge into the field. Prior to acceptance for publication, the following comments should be addressed:

Major concerns:
1. In ‘cell therapy procedure’ subsection, the authors did not have a proper control group. It should be processed as did a cell suspension for therapy.
2. It is well known that trypan blue is of reproductive toxicity. So the authors are strongly suggested to redo the experiments without use of trypan blue in cell therapy.
3. The authors used 5X10^4cells in cell injection. Why? Did the authors try other doses? If yes, what are the results?
4. Line 123, which the layer contained spermatogonial cells? The authors should clearly point out.
5. Figure 5 is wrongly presented. In the figure, the relative mRNA levels of vasa are shown, so the value of one group (for example sexually mature group) is required to set as 1±SE.
6. How many samples were used for gene expression analysis? (line 99)
7. The authors need to provide the evidence that the primers (actin and vasa) are suitable for qPCR analysis.
8. Line 154, there were 25 therapy-treated males and 5 control males. Why only 5 control males were used? The discrepancy of sample size is too big and the results are skeptical.

Minor concerns:
9. Too many figures. It is recommended to combine some related figures into one. Figure 1 and 8 are better to be deleted or added as supplementary figures.
10. Tables 1 and 2 should be reformed by three-line tables.
11. There are some English language errors. The authors need to double-check the MS.

Reviewer 3 ·

Basic reporting

Dear editor and authors
I reviewed a paper Resumption of donor-origin spermatogenesis in senescent goldfish Carassius auratus (Linnaeus, 1758) following spermatogonia cell therapy.
The authors confirmed that senescent fish are capable to restore spermatogenesis from transplanted spermatogonia. The topic and results are quite interesting as fundamental research and I can recommend the paper for publication after some revisions.
The main massage of the paper shouldn’t be misleading. It can be far applicable for aquaculture (line 73), since the germ cells are derived from sexually potent males and the spermatogenesis of germ cells from senescent fish itself was not restored.
Fig 6 F: Authors showed spermatozoa labelled by PKH26. Firstly, I don’t believe that the fluorescent dye can remain from spermatogonia after proliferation and differentiation until spermatozoa and secondly according to the scale bar, the spermatozoa should have about 400 µm in length, which is not possible. Goldfish spermatozoa has about ten times less.
Authors didn’t mention how many males were used for genotyping (blood and sperm from senescent fish after transplantation). Please provide the number.
Authors evaluate the age of senescent fish from scales, could they provide a picture to support this evaluation?

Experimental design

no commnet

Validity of the findings

no commnet

---

## Round 0.2 · Minor Revisions

Thank you very much for improving your work. Your manuscript still needs a minor change, You should draw the primers designing diagram to indicate how the primers distinguish the donor and recipient DNA.

Reviewer 1 ·

Basic reporting

No comments

Experimental design

No comments

Validity of the findings

No comments

Additional comments

The manuscript and figure were greatly improved according to the reviewer's comments. However,the author should draw the primers designing diagram to indicate how the primers distinguish the donor and recipient DNA. If the author could accomplish it, I suggest the manuscript to be accepted.

Reviewer 3 ·

Basic reporting

no comment

Experimental design

no comment

Validity of the findings

no comment

Additional comments

Authors have improved the manuscript according to the comments and therefore I can suggest it for publication.

---

## Round 0.3 · accepted · Accept

Dear Authors,

After revising the new version of your manuscript I am pleased to confirm that your paper has been accepted for publication in PeerJ.
Thank you for submitting your work to this journal.